# Temperature-Dependent Rheological and Viscoelastic Investigation of a Poly(2-methyl-2-oxazoline)-b-poly(2-*iso*-butyl-2-oxazoline)-b-poly(2-methyl-2-oxazoline)-Based Thermogelling Hydrogel

**DOI:** 10.3390/jfb10030036

**Published:** 2019-08-07

**Authors:** Michael M. Lübtow, Miroslav Mrlik, Lukas Hahn, Alexander Altmann, Matthias Beudert, Tessa Lühmann, Robert Luxenhofer

**Affiliations:** 1Polymer Functional Materials, Chair for Advanced Materials Synthesis, Department of Chemistry and Pharmacy and Bavarian Polymer Institute, Julius-Maximilians-University Würzburg, 97084 Würzburg, Germany; 2Centre of Polymer Systems, University Institute, Tomas Bata University in Zlin, Trida T. Bati 5678, 760 01 Zlin, Czech Republic; 3Institute of Pharmacy and Food Chemistry, Julius-Maximilians-University Würzburg, Am Hubland, 97074 Würzburg, Germany

**Keywords:** amphiphilic block copolymer, poly(2-oxazoline), viscoelasticity, thermoresponsive hydrogel, cytocompatibility

## Abstract

The synthesis and characterization of an ABA triblock copolymer based on hydrophilic poly(2-methyl-2-oxazoline) (pMeOx) blocks A and a modestly hydrophobic poly(2-*iso*-butyl-2-oxazoline) (p*i*BuOx) block B is described. Aqueous polymer solutions were prepared at different concentrations (1–20 wt %) and their thermogelling capability using visual observation was investigated at different temperatures ranging from 5 to 80 °C. As only a 20 wt % solution was found to undergo thermogelation, this concentration was investigated in more detail regarding its temperature-dependent viscoelastic profile utilizing various modes (strain or temperature sweep). The prepared hydrogels from this particular ABA triblock copolymer have interesting rheological and viscoelastic properties, such as reversible thermogelling and shear thinning, and may be used as bioink, which was supported by its very low cytotoxicity and initial printing experiments using the hydrogels. However, the soft character and low yield stress of the gels do not allow real 3D printing at this point.

## 1. Introduction

Thermoresponsive polymers are a type of “smart” material which change their appearance and physical properties upon a change in temperature [1]. Such materials can serve as functional biomaterials, for example, as bioinks [2]. The thermoresponsive character can manifest in various manners. First and most commonly, at a critical temperature, a clear polymer solution may become turbid as the polymer forms mesoglobules with strongly increased light scattering. This temperature is called the “cloud point”, and it may depend on the solvent quality and polymer concentration. The lowest cloud point with respect to the polymer concentration is called the lower critical solution temperature (LCST) and is often accompanied by a decrease in viscosity. This phenomenon is typically reversible but may show some hysteresis [3,4,5,6]. Second, thermoresponsive polymer solutions may undergo thermogelling, which can be described as a significant change in agglomeration/aggregation and which significantly increases the viscosity upon thermal stimulation [7,8,9]. In most cases, the transition from solution to gel takes place when the temperature increases [10,11,12]; however, there are also materials which show a gel-to-solution transition with increasing temperature, called “inverse gelation” [13]. One very prominent thermogelling polymer is Pluronic® F127, an ABA triblock copolymer bearing hydrophilic poly(ethylene glycol) as the A blocks and thermoresponsive poly(propylene glycol) as the B block. Another family of polymers which has a rich and versatile thermoresponsive profile and has received a lot of attention in recent years due to their applicability in various biomedical contexts are poly(2-oxazoline)s (POx) [3,14,15,16,17] These structural isomers of polypeptides are accessible via living cationic ring-opening polymerization (LCROP) of 5-membered cyclic imino ethers, the 2-oxazolines [18]. Besides the use of POx in, e.g., protein/drug conjugates [14,19,20,21,22,23,24,25], non-covalent drug delivery systems [26,27,28,29,30,31,32,33,34,35], and anti-fouling modifications/surfaces [36,37,38,39,40,41,42], a plethora of POx-based hydrogels as soft biomaterials have been developed [43,44,45,46]. The emergence of the latter was supported by the low cytotoxicity/high biocompatibility of POx-based hydrogels [43], enabling their application as, e.g., bioink [46].

In a recent study [47], a small library of POx-based ABA triblock copolymers were synthesized based on 2-*n*-propyl-2-oxazoline and 2-methyl-2-oxazoline with the aim to mimic the behavior of Pluronic® F127. However, these polymers did not show any gelation, even at higher concentration (20–30 wt %) even though the materials exhibit cloud points depending on the ratio between individual blocks. In contrast, Monnery and Hoogenboom very recently reported that a BAB triblock copolymer with an inner block of poly(2-ethyl-2-oxazoline) and outer blocks of poly(2-*n*-propyl-2-oxazoline) showed irreversible thermal gelation at 20 wt % upon heating if the central block showed an extremely high degree of polymerization of 900 [48]. Contrary to this, diblock copolymers bearing poly(2-n-propyl-2-oxazine) and poly(2-methyl-2-oxazoline) blocks showed reproducible thermal gelation at concentrations above 20 wt % at much lower total degrees of polymerization [46]. The obtained storage modulus for this system, containing 100 monomeric units of each block, was found to be 4 kPa.

Here, we investigated the viscoelastic properties of aqueous solutions of a POx-based, amphiphilic ABA triblock copolymer comprising two hydrophilic poly(2-methyl-2-oxazoline) (pMeOx) blocks A and a modestly hydrophobic poly(2-*iso*-butyl-2-oxazoline) (p*i*BuOx) block B (A-p*i*BuOx-A). A copolymer with the same structure (but with slightly different chain lengths and a different end group) was previously investigated as a drug-delivery system for non-water-soluble drugs such as taxanes [49], curcumin, efavirenz, and others [50]. However, in this study, thermogelling behavior was not observed. 

## 2. Materials and Methods

### 2.1. Materials

All substances for the preparation of the polymer were purchased from Sigma-Aldrich (Steinheim, Germany) or Acros (Geel, Belgium) and were used as received unless otherwise stated. The monomer 2-*iso*-butyl-2-oxazoline (*i*BuOx) was prepared as previously described [49]. Deuterated chloroform for NMR analysis was obtained from Deutero GmbH (Kastellaun, Germany). The substances used for polymerization, specifically methyl trifluoromethylsulfonate (MeOTf), 2-methyl-2-oxazoline (MeOx), and 2-*iso*-butyl-2-oxazoline (*i*BuOx), were refluxed over CaH_2_, distilled, and stored under argon. Benzonitrile (PhCN) was refluxed over phosphorus pentoxide, distilled, and stored under argon. For a detailed description of *i*BuOx synthesis and characterization, the reader is referred to Appendix A in the Appendix A.

### 2.2. Polymer Synthesis

The polymerization and work-up procedures were carried out as described previously [26]. For polymer synthesis and characterization, see Appendix A. Briefly, initiator was added to a dried and nitrogen-flushed flask and dissolved in PhCN. The monomer 2-methyl-2-oxazoline (MeOx) was added, and the reaction mixture was heated to 100 °C for approximately 4 h. Reaction progress and monomer consumption was controlled by FTIR and ^1^H-NMR spectroscopy. After complete consumption of MeOx, the mixture was cooled to RT, and 2-*iso*-butyl-2-oxazoline was added. The reaction mixture was heated to 100 °C overnight. The procedure was repeated for the third block, MeOx, and after monomer consumption was confirmed, termination was carried out by addition of 1-tert-butyl piperazine 1-carboxylate (PipBoc) at 50 °C for 4 h. Subsequently, K_2_CO_3_ was added and the mixture was stirred at 50 °C for 4 h. Precipitates were removed by centrifugation and the solvent removed under reduced pressure. The supernatant was transferred into a dialysis bag (molecular weight cut-off (MWCO) 1 kDa, cellulose acetate) and dialyzed against deionized water overnight. The solution was recovered from the bag and lyophilized. 

## 3. Methods

### 3.1. Nuclear Magnetic Resonance Spectroscopy (NMR)

NMR spectra were recorded on a Fourier 300 (^1^H: 300.12 MHz), Bruker Biospin (Rheinstetten, Germany) at 298 K. The spectra were calibrated to the signal of residual protonated solvent (CDCl_3_ at 7.26 ppm).

### 3.2. Gel Permeation Chromatography (GPC)

Gel permeation chromatography (GPC) was performed on an Agilent 1260 Infinity System (Polymer Standards Service (PSS), Mainz, Germany) with hexafluoroisopropanol (HFIP) containing 3 g/L potassium trifluoroacetate as eluent; precolumn: 50 × 8 mm PSS PFG linear M; two columns: 300 × 8 mm PSS PFG linear M (particle size 7 µm; pore size 0.1–1000 kg/mol). The columns were kept at 40 °C and the flow rate was 0.7 mL/min. Prior to each measurement, samples were filtered through 0.2 μm PTFE filters (Roth, Karlsruhe, Germany). Conventional calibration was performed with PEG standards (0.1–1000 kg/mol), and data were processed using WinGPC software. 

### 3.3. Characterization of the Hydrogel

The polymer solution was prepared by dissolving A-p*i*BuOx-A at 20 wt % in deionized water For complete dissolution, the solution was shaken at 4 °C overnight.

### 3.4. Rheological Investigations

Rheological and viscoelastic properties were investigated using the Rheometer Physica MCR-301 (Anton Paar, Graz, Austria). Measurements were performed using a stainless steel parallel plate geometry with 25 mm (PP 25) in diameter and a 300 μm gap between the upper and lower plates in all investigations. For steady shear experiments, the control shear rate mode was used to investigate the rheological behavior from 1 s^−1^ to 500 s^−1^, and 6 points per decade were used with a logarithmic increase and at various temperatures (5 °C, 25 °C, and 37 °C). In order to investigate the reversibility of the gelation created in the system upon elevated temperatures, the viscosity data were obtained at a shear rate of 1 s^−1^, and a discrete, stepwise increase of temperatures from 5 °C to 25 °C to 37 °C and back was performed three times. For equilibration, the system was given a one-minute waiting time to reach the desired temperature for both the heating and cooling regimes. The steady shear rheological data, namely, the dependence of the shear stress on the shear rate, were plotted using the Vocadlo model shown in Equation (1) [51], and parameters of the model were calculated using a least square method:
(1)$$\tau =[K^{\frac{1}{n}}{\left|\overset{\cdot }{\gamma }\right|}^{\frac{n-1}{n}}+(\frac{\tau_{0}}{\left|\overset{\cdot }{\gamma }\right|}{)}^{\frac{1}{n}}{]}^{n}\;\overset{\cdot }{\gamma }$$
where *τ* is the shear stress, *K* is the consistency index, *n* is the power law index [51] or flow index [52], $\overset{\cdot }{\gamma }$ is the shear rate, and *τ*_0_ is the yield stress as a measure of the internal structural integrity developed upon elevated temperature.

According to the literature, the Herschel–Bulkley (H-B) model is recommended for investigation of the yield stress for hydrogel structures [52]. The H-B model is described in Equation (2), and the parameters have the same meaning as was described for the Vocadlo model previously:
(2)$$\tau =\tau_{0}+K\cdot {\overset{\cdot }{\gamma }}^{n}.$$

For the investigation of the viscoelastic properties, all measurements were performed in the linear viscoelastic range (LVR). This range was obtained from the amplitude sweep where the storage G′ and loss moduli G″ were measured against the strain deformation from 0.01% to 10% for A-p*i*BuOx-A at 5 °C and from 0.01% to 100% at 25 °C and 37 °C. From these experiments, the yield stress was also determined by plotting the dependency of storage G′ and loss moduli G″ on the applied shear stress. The onset of G′ decrease was defined as the yield point. Then, the frequency sweep from 0.1 Hz up to 100 Hz was measured at the obtained deformation from LVR, namely 2% for 5 °C, 0.2% for 25 °C, and 0.3% for 37 °C. Moreover, the viscoelastic behavior was investigated at 25 °C and 37 °C at a frequency of 1 Hz, and deformations were alternated from 0.1% to 10%. Here, five cycles were measured. This measurement was not performed at 5 °C due to the liquid-like behaviour of the sample; thus, just small differences would be expected due to the nearly Newtonian behavior. Finally, the temperature sweep from 5 °C to 45 °C was measured at a strain of 1% and a frequency of 1 Hz in order to prove the reversible sol-gel transition upon temperature sweep measurement.

### 3.5. Microscopic Investigations

Microscopic examination was performed on a Crossbeam 340 scanning electron microscope (Zeiss, Oberkochen, Germany). Aqueous polymer solution (20 wt %) was heated to 40 °C (gel state), shock frozen in liquid nitrogen, and freeze-dried. Prior to the measurement, the freeze-dried sample was sputter coated with 20 Å of platinum using an EM ACE 600 sputter coater (Leica, Wetzlar, Germany).

### 3.6. Printing

Hydrogel scaffolds of a 20 wt % aqueous solution were printed at 30 °C using a compact bench-top 3D bioprinter (Incredible, Cellink, Gothenburg, Sweden) working on the principle of a pneumatic extrusion-based printer. The printing speed was set to 600 mm/min, and a pressure of 50 kPa was applied using a 0.2 mm (27 GA) diameter nozzle (8 mm length) (Nordson EFD, Oberhaching, Germany). First, the printability was established by printing one layer with different strand distances varying over 0.75 mm, 1 mm, 1.25 mm, and 1.5 mm. Two- and four-layered constructs of 12 × 12 mm with four strands and a layer height of 0.3 mm were printed.

### 3.7. Cell Culture

Human embryonic kidney HEK293 cells (HEK-Blue^TM^ IFN-α/β, Invivogen) were cultured in growth medium (DMEM 10% fetal calf serum (FCS), 100 U/mL penicillin G, and 100 µg/mL streptomycin) on 25 cm^2^ culture flasks at 37 °C and 5% CO_2_. Calu-3 cells (human lung adenocarcinoma, ATCC HTB-55) were maintained in growth medium (MEM 10% FCS, 100 U/mL penicillin G, 100 µg/mL streptomycin, 1% non-essential amino acids (NEA), 1 mM pyruvate, 2 mM glutamine, and 2.88 g/L glucose) at 37 °C and 5% CO_2_. 

### 3.8. WST-1 Proliferation Assay

For the measurement of the cytotoxicity of the polymer, 5000 cells/well of both Calu-3 and HEK cells were seeded in 96-well plates in growth medium and incubated at 37 °C and 5% CO_2_ for 24 h and 48 h, respectively. Final polymer concentrations of 10%, 5%, 1%, and 0.1% were prepared from a stock solution (20 wt %) in growth medium on ice and added to the cells. After 24 h of cell growth, the medium was removed and replaced by fresh cell culture medium. Cells were incubated with WST-1 reagent for 1–4 h at 37 °C according to the manufacturer′s manual. The formation of formazan was monitored at 450 and 630 nm using a Spectramax 250 microplate reader (Molecular Devices, San Jose, CA, USA). Results were normalized using cells incubated with polymer-free cell culture medium.

## 4. Results and Discussions

Previously, A-p*i*BuOx-A was utilized for the solubilization of hydrophobic molecules at a concentration of 10 g/L. Its critical micelle concentration, however, lies much lower, at around 12 mg/L [50]. Inspired by the ongoing search for novel bioinks for utilization in biofabrication [53] and by our previous work on thermoresponsive di- and triblocks based on poly(2-oxazoline)s and poly(2-oxazine)s [46,47], we tested the visual appearance of higher concentrations of aqueous solutions of A-p*i*BuOx-A up to 20 wt % (Figure 1) at different temperatures ranging from 5 °C to 80 °C. At all concentrations and temperatures, the solutions were turbid, more so with increasing concentration, indicating the presence of larger self-assemblies or mesoglobules scattering light. At 5 °C, all samples were clearly liquids of rather low viscosity that flowed freely. At higher temperatures, the viscosity of the 20 wt % solution visibly increased until a turbid gel was formed at 20 °C. The visual impression remained unchanged for all samples up to 80 °C. It is important to note that neither of the blocks of the triblock copolymer exhibited an LCST at around 20 °C. While p*i*BuOx is essentially water-insoluble even at around 0 °C, pMeOx is excellently water-soluble even at temperatures approaching 100 °C. Therefore, the gelation cannot be attributed to an LCST of either block, but is a property of the triblock, similar to the situations observed for other thermogelling triblock copolymers [54].

A first impression of the microscopic structure of the hydrogel was obtained by scanning electron microscopy of freeze-dried samples. The structure of the prepared hydrogel is crucial from the point of view of its potential applicability, since the porosity and network structure of the hydrogel influence its mechanical properties. It can be seen that the sample exhibited a porous structure with open pores and a pore size in the range of several micrometers (Figure 2). Such big pores are not necessarily expected for physical hydrogels at such high concentrations, and we cannot rule out at this point that the observed structure may in part be an artefact from sample preparation (freeze-drying). In particular, some features such as the sharp, teeth-like protrusions (Figure 2, right-hand image) are likely to have originated from sample preparation.

In order to investigate the reversibility and repeatability of the internal structure formation due to physical cross-linking, the viscosity profiles of the 20 wt % sample were repeatedly recorded during three heating and cooling cycles at 5 °C, 25 °C, and 37 °C (Figure 3a). At 5 °C, viscosities of only 2 Pa∙s were obtained. Changing the temperature to 25 °C, a rapid increase in viscosity to 35 Pa∙s occurred, further increasing to 60 Pa∙s at 37 °C. This pronounced increase in viscosity highlights the material’s tunable properties upon various temperature profiles. Most importantly, this behavior was reversible, as cooling down the samples from 37 °C to 25 °C decreased the viscosity again. However, a small hysteresis was observed for the cooling step to 25 °C, as slightly higher viscosities were obtained than at the initial heating stage at 25 °C. In contrast, the initial values of 2 Pa∙s were immediately reached upon cooling to 5 °C. During the three cycles, the viscosity profiles upon temperature change were essentially identical.

To determine the reproducibility of the liquefaction and gelation upon continuous temperature change, the viscoelastic properties of the polymer gel were investigated in eight heating/cooling cycles between 5 °C and 45 °C (Figure 3b) in oscillation mode. Again, the results were highly reproducible. The gel points determined as the crossover point of the storage and loss moduli always occurred at approximately 22 °C. Furthermore, the liquefaction process was also very similar during all cycles. Only a minor increase in the storage moduli at low temperatures occurred with progressing cycles, which might be due to minor dehydration of the samples at higher temperatures (45 °C). However, such an increase in G′ was not observed at higher temperatures, which does not corroborate significant dehydration. Clearly, the gelation and liquefaction process was highly reproducible and reversible for this hydrogel; this is in contrast to recently published hydrogels based on POx BAB triblock copolymers, which exhibited higher values of the storage modulus but showed irreversible crosslinking (gelation) in consecutive heating/cooling cycles [48].

In order to investigate the gelation process in more detail, the viscoelastic properties were determined. At 5 °C, no relevant data at low strain deformations of <0.4% could be obtained due to the low viscosity (Figure 4a). At strain deformations between 0.4% and 10%, the liquid-like behavior of the polymer solution was confirmed, as the loss modulus G″ exceeded the storage modulus G′. Increasing the temperature to 25 °C led to markedly altered viscoelastic properties. At low deformations, the storage modulus dominated, indicating that physical cross-linking was already present. With increasing strain deformation, shear thinning occurred, resulting in liquid-like properties above 1% deformation. A further increase in temperature to 37 °C strengthened the physical cross-links. This was represented by higher initial G′-values as well as longer-lasting solid-like behavior up to deformations of 3%. Similarly, as for 25 °C, significant recovery upon deformation was observed, which will be investigated in more detail below. 

The frequency dependence of the viscoelastic moduli exhibited a marked temperature dependence, as the initial liquid-like behavior of the sample at 5 °C vanished with increasing frequency, resulting in solid-like properties at frequencies of >2.5 Hz (Figure 4b). Such behavior is common for similar types of polymer solutions [54]. However, at 25 °C and 37 °C, the frequency had only a minor effect on the viscoelastic properties of the gel. The solid-like behavior remained during the whole frequency range up to 100 Hz. At 25 °C, the underlying physical cross-linking network was still affected by the frequency, as noticeable fluctuation of the storage modulus occurred. At 37 °C, it appears that well-developed physical cross-links with storage moduli around 700 Pa were stable in the whole frequency range. 

To obtain the values of the yield stress, the viscoelastic moduli were plotted against the shear stress (Figure 4c). The yield stress was evaluated by tangent analysis of the drop of the storage modulus from the linear region. It can be seen that the sample measured at 5 °C had quite scattered data due to the low viscosity, similar to that seen in Figure 4a, and was not evaluated. Increasing the temperature to 25 °C yielded somewhat better data; however, the storage modulus was not entirely constant at low shear stress and decreased more gradually. Nevertheless, a tentative yield stress of 2 Pa could be obtained. The more pronounced physical cross-linking at 37 °C allowed better analysis and resulted in an enhanced value for the yield stress of 20 Pa. These findings are in good agreement with investigations of the yield stress from steady shear measurements, which are described in the following.

For an alternative determination of the yield point, steady shear measurements were performed. At 5 °C, the viscosity decreased very slowly with increasing shear rate indicating mostly Newtonian or slightly pseudo-plastic behavior (Figure 5a). Increasing the temperature to 25 °C increased the viscosity by over one order of magnitude, again indicating internal structure formation. Furthermore, more pseudo-plastic behavior was observed, as the decrease in viscosity with increasing shear rate was more pronounced at elevated temperature. Finally, increasing the temperature to 37 °C further increased the viscosity, corroborating intensified physical cross-linking and strengthening of the gel structure while retaining a pronounced shear thinning character. 

The viscosity decreased by more than two orders of magnitude at high shear rates. Such behavior can be advantageous from a practical point of view, as the material can be, e.g., injected quite easily, but it might provide sufficiently high viscosity for local embolization.

In order to quantify the yield stress of the polymer gel formed at elevated temperatures, the experimental data (Figure 5b) were fitted according to the Vocadlo Equation (1) and Herschel–Bulkley model to calculate the yield stress, consistency index, and index of non-Newtonian behavior (Table 1).

At 5 °C, both fits provided a very low yield stress and a power law index *n* close to 1, indicating only slightly pseudo-plastic behavior of the polymer solution. In contrast, much higher yield stresses were obtained at 25 °C and 37 °C. The fit using the Vocadlo model did not provide reasonable values for the power law indices and consistency indices for samples measured at 25 °C and 37 °C. In contrast, the Herschel–Bulkley model provided values for the consistency index, power law index, and yield stress that appear more reasonable. The power law index decreased with increasing temperatures, confirming the transition from liquid-like to solid-like state and pronounced shear thinning behavior. Moreover, the consistency index increased, as did the yield stress, as a result of the created physical cross-links in the material at elevated temperature, which is in agreement with the evaluated values during the dynamic oscillation stress/strain sweeps. The obtained yield stress of 36 Pa at 37 °C is rather low in comparison to common hydrogels mentioned in a review article by Townsend et al. [52]; however, it is still higher than values reported for other materials elsewhere [55,56,57].

The experimentally determined value of the yield stress for the hydrogel measured at 37 °C (Figure 4c) is in the same order of magnitude but slightly lower compared to the calculated yield stress from the H-B model. This difference is tentatively attributed to wall slippage present between the hydrogel and stainless steel geometry.

In order to investigate the shear thinning and structure recovery properties in more depth, the viscoelastic moduli where collected over time at two regimes with either 0.1% or 10% strain deformation (Figure 6). For this, the sample at 5 °C was omitted due to the negligible change between these two strain deformations. At low deformation, both samples measured at 25 °C and 37 °C exhibited strong, solid-like behavior, as the storage modulus significantly dominated over the loss modulus.

However, at 10% strain deformation, both samples dramatically changed to liquid-like behavior with a dominating loss modulus, similarly to what was observed for a hydrogel based on hyaluronic acid, albeit at much higher strain deformation (500%) [58]. This change was also confirmed by tan *δ* visualization at various deformation regimes: at low deformations (0.1%) the samples exhibited tan *δ* = 0.25 and 0.18, while for high deformations (10%) they exhibited tan *δ* = 1.5 and 1.8 (for 25 °C and 37 °C, respectively). Moreover, this change in viscoelastic properties was fully reversible during five deformation cycles. However, after the high strain regime, the return to full strength took a significant amount of time (approximately 2 min) even though return to the gel state appeared to be immediate. This observation stands in stark contrast to the rapid recovery to full strength of a previously described POx-based bioink [44]. This again is a good foundation for possible injection of the polymer gel into the body as, e.g., an embolization agent or drug depot, since upon small deformations, solid-like behavior occurred. However, the slow recovery of the gel strength in combination with the low yield stress might prove problematic with respect to shape fidelity in 3D printing.

Cytocompatibility is an important property regarding applications, e.g., as bioink [46] or drug depots. Therefore, we investigated the cytotoxicity of different non-gelling concentrations ranging from 0.1 to 10 wt % and observed no dose-dependent cytotoxicity in either HEK293 and human Calu-3 cells after 24 h of incubation (Figure 7). However, we did see an increase in the apparent cell viability. This has been observed before for POx-based polymers in select cases but not as pronounced in the case of Calu-3 cells observed here [35]. This phenomenon requires further and detailed investigations; these are, however, outside the scope of the present contribution.

For the presented physically cross-linked hydrogel, one potential application could be as a functional printable ink. Therefore, hydrogel scaffolds of a 20 wt % aqueous solution were printed at 30 °C using a compact extrusion-based bench-top 3D bioprinter with a printing speed of 600 mm/min utilizing 50 kPa constant pressure. First, the printability was tested by printing one layer with different strand distances (Figure 8a). It is apparent that at the largest distance tested (1.5 mm), some strand separation can be observed, while at the lowest distance (0.75 mm), the printed strands merged into one homogenous layer. However, the layer deposition is rather exact and deposited strands remained well in position (Figure 8b). Subsequently, two- and four-layered constructs of 12 × 12 mm with four strands (3 mm layer distances) and a layer height of 0.3 mm were printed. It is clearly evident that layers printed on top of each other merge, so real 3D printing is not possible at this point with this material. This can also readily be explained by the comparably low viscosity and storage moduli values in the gel state in conjunction with the low yield stress of the material. While in the case of the two-layered print, the overall 4 × 4 grid remains reasonably well resolved (Figure 8c), the strands fuse significantly when four strands are printed on top of each other, showing that the material is obviously not able to hold its own weight—a fundamental prerequisite for real 3D printing. However, future work will address whether modifications in the polymer structure or additives such as laponite may be useful to improve the printing properties of the presented hydrogel, as has been previously reported by Peak et al. [59] and others. It is important to note that if the shape fidelity can be improved, cytocompatibility must again be assessed and conditions relevant for printing will also have to be tested.

## 5. Conclusions

In this study, an ABA triblock copolymer based on poly(2-methyl-2-oxazoline) and poly(2-*iso*-butyl-2-oxazoline) was synthesized and characterized by ^1^H-NMR and GPC. Visual inspection of polymer solutions at various concentrations and temperatures revealed thermal gelation at concentrations of at least 20 wt % of the ABA triblock copolymer in water. Steady shear experiments at various temperatures confirmed the transition from solution to a gel-like structure. The viscoelastic investigation showed a soft hydrogel forming above 25 °C exhibiting an elastic modulus of 150 Pa, while at 37 °C it showed an elastic modulus of almost 600 Pa. Furthermore, shear thinning behavior was confirmed to be highly reversible, as the elastic moduli measured at 37 °C changed from 600 Pa to 20 Pa for 0.1% and 10% strain deformation, respectively. Finally, the reproducibility of the thermal gelation was confirmed using eight cycles with transition from 5 °C to 37 °C. The material exhibited low cytotoxicity in both Calu-3 and HEK cells at all investigated polymer concentrations after 24 h at 37 °C. The results of the presented rheological, viscoelastic, and printing investigations suggest that the fabricated hydrogel may be used as a functional printable bioink, but further modifications will be necessary to improve shape fidelity. This could include light-induced curing of polymers modified with cross-linking functionalities, addition of viscoelastic modifiers such as laponite or further tuning of the polymer structure. 

## Figures and Tables

**Figure 1 jfb-10-00036-f001:**
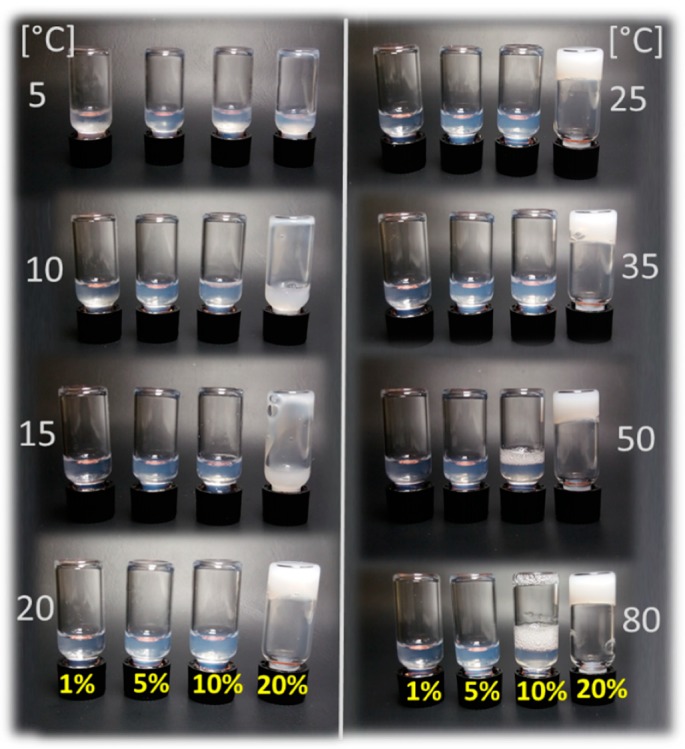
Visual appearance and flow properties of A-p*i*BuOx-A dissolved in H_2_O at 1, 5, 10, and 20 wt % (left to right vials) at 5–80 °C. Only at 20 wt %, a turbid, non-flowing gel was formed at T > 20 °C (highly viscous sol at T = 15 °C).

**Figure 2 jfb-10-00036-f002:**
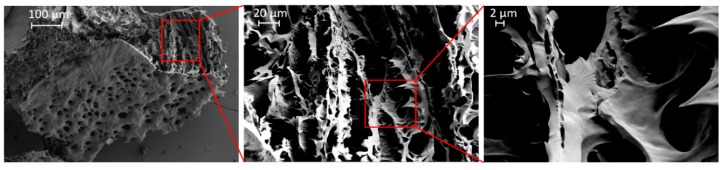
SEM images of the freeze-dried hydrogel (20 wt %) at different magnifications displaying a porous surface and a 3D-interconnected interior.

**Figure 3 jfb-10-00036-f003:**
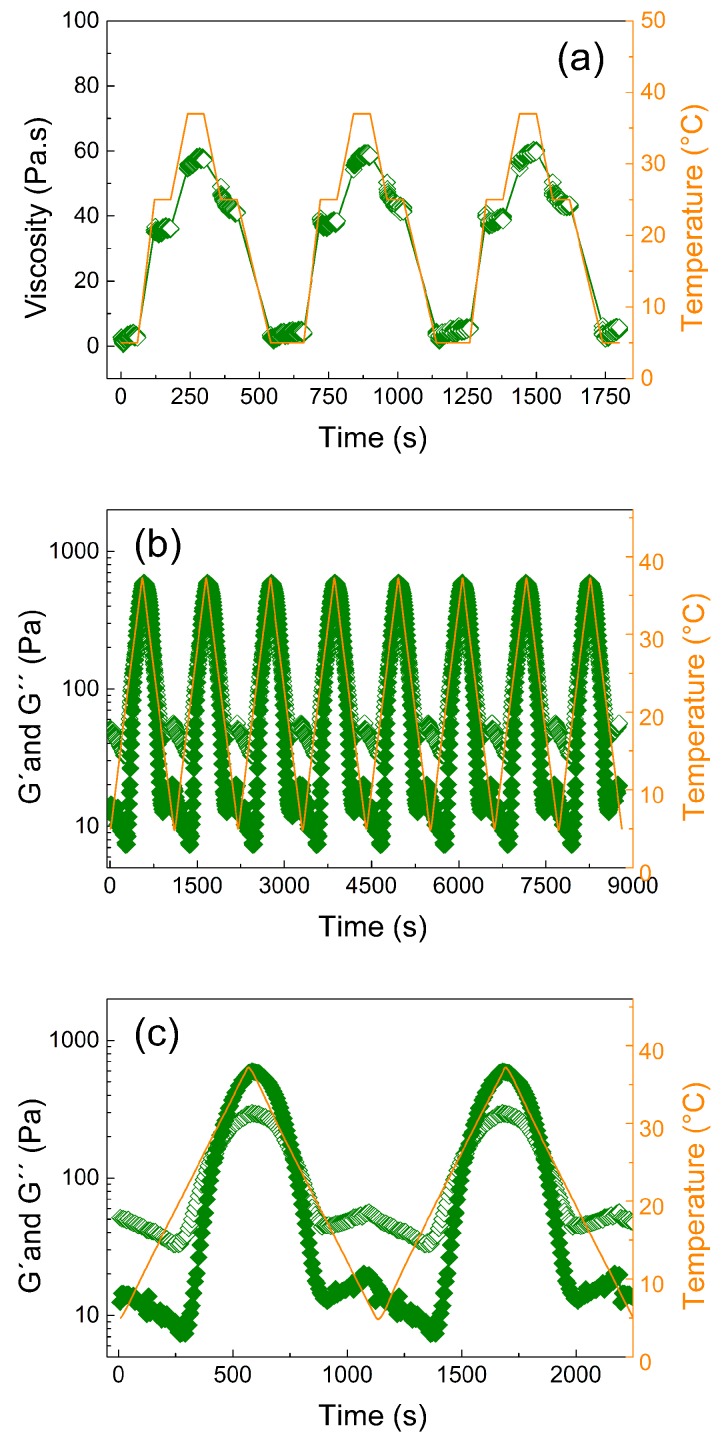
(**a**) Viscosity measurement under steady shear conditions over time for 20 wt % polymer solution at different temperatures (5 °C, 25 °C, and 37 °C) and a constant shear rate, *g*, of 1 s^−1^. (**b**) Dependence of the storage, G′, (solid symbols) and loss modulus, G″, (open symbols) on time at simultaneously increasing and decreasing temperature (5 °C and 37 °C) using eight heating/cooling cycles with a rate of 3 °C min^−1^ at 1% strain deformation and frequency 1 Hz. (**c**) First two cycles from Figure 3b to more clearly show the difference between the storage and loss moduli during cycling.

**Figure 4 jfb-10-00036-f004:**
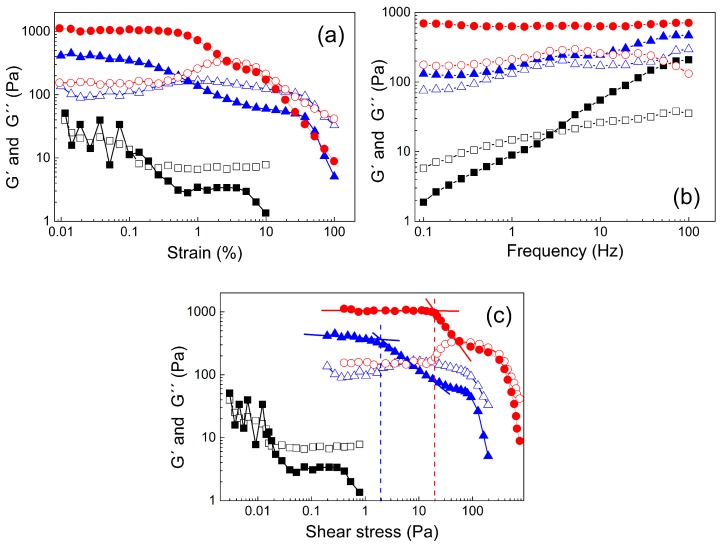
Dependence of the storage G′ (solid symbols) and loss modulus G″ (open symbols) on the strain deformation (**a**), frequency (**b**), and shear stress (**c**) for 20 wt % polymer solution at various temperatures (■, 5 °C; ▲, 25 °C; ●, 37 °C).

**Figure 5 jfb-10-00036-f005:**
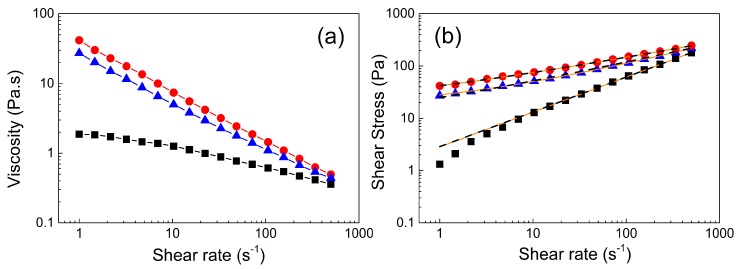
Dependence of the viscosity (**a**) and shear stress (**b**) on the shear rate at various temperatures (black square ■, 5 °C; blue triangle ▲, 25 °C; and red circle ●, 37 °C). Orange solid lines in Figure 5b represent the Vocadlo model fit and black dashed lines represent the Herschel–Bulkley (H-B) model fit.

**Figure 6 jfb-10-00036-f006:**
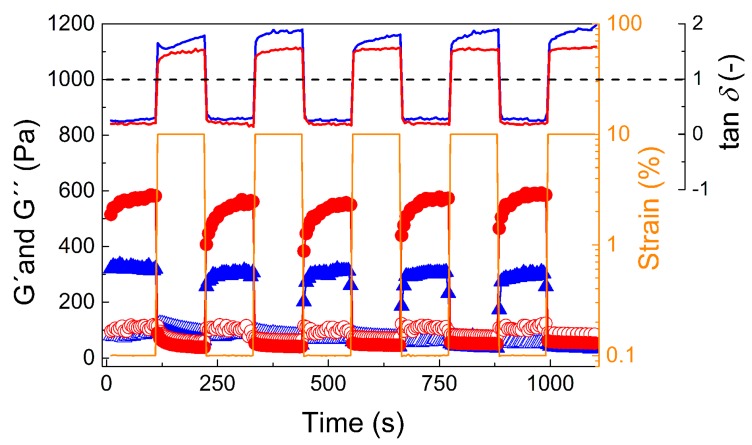
Dependence of the storage, G′ (solid symbols), and loss modulus, G″ (open symbols), on time at various strain deformation regimes (0.1% and 10%) for 20 wt % polymer solution. Samples were measured at 25 °C (blue triangles) and 37 °C (red circles) and at frequency 1 Hz. Solid lines represent the values of tan *δ* for hydrogel samples measured at 25 °C (blue solid line) and 37 °C (red solid line). The black dashed line represents tan *δ* = 1 as a guide for the eye to differentiate between the solid-like and liquid-like states at various strain deformations.

**Figure 7 jfb-10-00036-f007:**
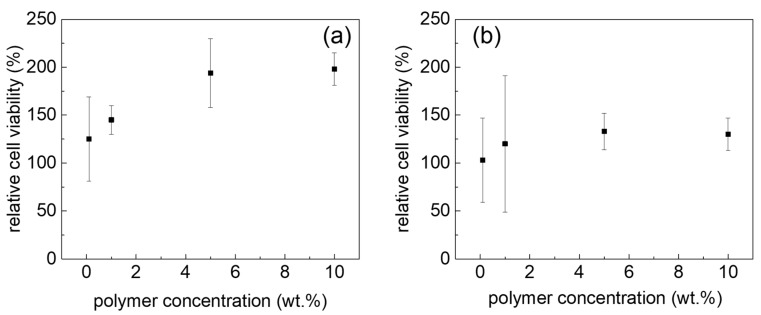
Relative cell viability of Calu-3 (**a**) and HEK cells (**b**) at different block copolymer concentrations after 24 h incubation at 37 °C, normalized to the cell viability of Calu-3 and HEK cells cultured without polymer. Values are presented as means ± standard deviations (*n* = 3).

**Figure 8 jfb-10-00036-f008:**
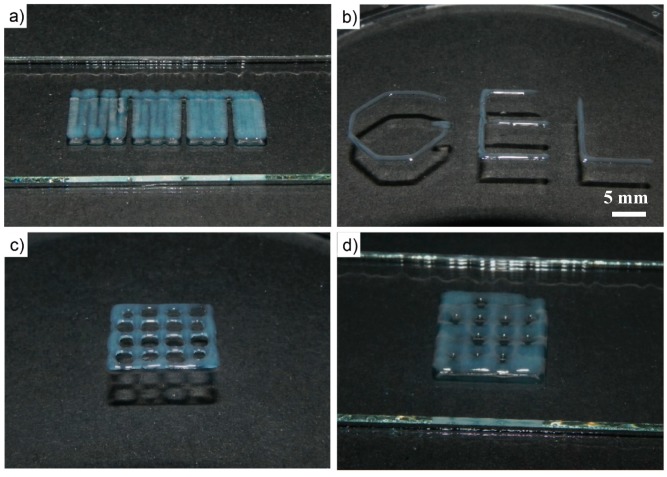
Photographic images of printing experiments using 20 wt % A-p*i*BuOx-A aqueous solution. (**a**) Using variable strand distances (0.75 mm–1.5 mm in 0.25 mm steps), merging of adjacent strands can be assessed. (**b**) Strand placement is reasonably exact and placed strands stay in place well. Photographic images of (**c**) two- and (**d**) four-layered printed constructs of 12 × 12 mm with four strands and a layer height of 0.3 mm.

**Table 1 jfb-10-00036-t001:** Summarized parameters obtained from the calculation of the Vocadlo and Herschel–Bulkley models.

Temperature [°C]	*τ*_0_, Yield Stress [Pa]	*K*, Consistency [Pa∙s]	*n*, Non-NEWTONIAN Index [-]
Vocadlo model parameters			
5 °C	0.05	0.16	0.89
25 °C	70	0.20	0.70
37 °C	117	0.22	0.69
Herschel–Bulkley model parameters			
5 °C	0.05	2.8	0.77
25 °C	22	18	0.39
37 °C	36	28	0.34

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
