# Peer review of "Temperature-Dependent Rheological and Viscoelastic Investigation of a Poly(2-methyl-2-oxazoline)-b-poly(2-iso-butyl-2-oxazoline)-b-poly(2-methyl-2-oxazoline)-Based Thermogelling Hydrogel"

_jfb, 2019, doi:10.3390/jfb10030036_

Round 1

Reviewer 1 Report

Report on a “temperature dependent rheological and viscoelastic investigation of aqueous solutions of poly(2-methyl-2oxazoline)-b(poly(2-iso-butyl-2-oxazoline)-b-poly(2-methyl-2-oxazoline), a novel thermogelling hydrogel”

Lübtow et al present in this paper a study about a novel thermogelling hydrogel, a ABA triblock copolymer based on hydrophilic poly(2-methyl-2oxazoline) block and a hydrophobic poly(2-iso-butyl-2-oxazoline) block.

I recommend the acceptance of the article after the following comments have been taken into account.

In the introduction, the authors exposed clearly the interests of POx based hydrogel, especially ABA triblock copolymer in order to use them as bioinks. But Monnery et al have just published a study in Polymer Chemistry (2019, DOI:10.1039/C9PY00300B) on this kind of triblock copolymer with the same A block but the poly(2-n-propyl-2oxazoline) for the block B. They should be cited by the authors. They should compare their results to this paper. The difference of the studies is the complete rheological study and the application of 3D printing in these article presented by Lübtow et al.

Comments:

Page 2, Line 75: reference [47] should be after the period.

In the materials and method paragraph, the authors are referred to supporting information, but I didn’t have access to supporting information. So the figures which are necessary should be included in this paper.

Figure 3 gives the dependence of the storage G’ and the loss modulus G’’ on the time at cycle of temperature. But it is very difficult to see the difference between the solid symbol and the open symbol. It should be better and easier to understand to represent the tan δ instead of G’ and G’’. Tan d stand for the ration between G’’/G’. If G’’>G’, tanδ >1 the sample is a liquid like, if tanδ<1 the sample is solid like.

Figure 6 represents the dependence of G’ and G’’ versus time as the figure 3, the scale of G’ and G’’ should be logarithm scale instead of linear scale. tanδ can be used as previously.

Author Response

Reviewer #1

Report on a “temperature dependent rheological and viscoelastic investigation of aqueous solutions of poly(2-methyl-2oxazoline)-b(poly(2-iso-butyl-2-oxazoline)-b-poly(2-methyl-2-oxazoline), a novel thermogelling hydrogel”

Lübtow et al present in this paper a study about a novel thermogelling hydrogel, a ABA triblock copolymer based on hydrophilic poly(2-methyl-2oxazoline) block and a hydrophobic poly(2-iso-butyl-2-oxazoline) block.

I recommend the acceptance of the article after the following comments have been taken into account.

In the introduction, the authors exposed clearly the interests of POx based hydrogel, especially ABA triblock copolymer in order to use them as bioinks. But Monnery et al have just published a study in Polymer Chemistry (2019, DOI:10.1039/C9PY00300B) on this kind of triblock copolymer with the same A block but the poly(2-n-propyl-2oxazoline) for the block B. They should be cited by the authors. They should compare their results to this paper. The difference of the studies is the complete rheological study and the application of 3D printing in these article presented by Lübtow et al.

Response:We thank the reviewer for this valuable comment, indeed we were aware of this work, which became available only after submission of our manuscript. The requested reference has been added to the introduction section and also to the discussion section and corresponding comments are add to the revised version of the manuscript.

Comments:

Page 2, Line 75: reference [47] should be after the period.

Response: We thank the reviewer for this comment. The authors believe that the position of the reference is on appropriate place, mainly due to the fact that rest of the paragraph is more focused on the description of the commercial materials rather than synthesis of the triblock copolymer.

In the materials and method paragraph, the authors are referred to supporting information, but I didn’t have access to supporting information. So the figures which are necessary should be included in this paper.

Response: We thank the reviewer for this important comment. We were quite sure that we also submitted the extensive supporting information with our first submission. If that was not the case, we apologize and of course will provide this upon submission of the revised manuscript. 

Figure 3 gives the dependence of the storage G’ and the loss modulus G’’ on the time at cycle of temperature. But it is very difficult to see the difference between the solid symbol and the open symbol. It should be better and easier to understand to represent the tan δ instead of G’ and G’’. Tan d stand for the ration between G’’/G’. If G’’>G’, tanδ >1 the sample is a liquid like, if tanδ<1 the sample is solid like.

Response: We thank the reviewer for this critical comment. We showed the overview in Fig. 3b in order to provide an information about reproducibility of the phenomenon and therefore chose to present eight consecutive temperature cycles. However, we agree that symbols of storage and loss moduli cannot be differentiated properly. Therefore, we added Fig. 3c to the revised version of the manuscript showing the section of the first two cycles, whereby the changing of the moduli upon heating/cooling cycles is more clearly visualized.

Figure 6 represents the dependence of G’ and G’’ versus time as the figure 3, the scale of G’ and G’’ should be logarithm scale instead of linear scale. tanδ can be used as previously.

Response: We thank the reviewer for this comment but kindly disagree. There is an important difference between Figures 3 and 6. While in Figure 3, the temperature is changed (cycled) over time, the strain is cycled in Figure 6. In the latter case, the linear scale was used to better visualized the difference at various strain deformations over time. Logarithmic scale is not necessary and could in fact be misleading. However, the visualization of tan δ at various strain modes was added to Fig. 6, please see the revised version of the manuscript.

Reviewer 2 Report

The current manuscript describes an investigation of the synthesis of a novel thermogelling hydrogel and characterization of properties relevant for the application as a bioink. The authors indicate 4 variations of the mixture including 1, 5, 10, and 20% weight by volume, and further characterized the 20 wt%.  

Overall, the investigation has indicated multiple appropriate characterizations to analyze the proposed application.  However, the level of development in methodology and presentation of results is lacking. 

Examples for the level of detail missing in methods includes the following:

1. H-NMR and GPC mentioned performed in methods and discussion, results never presented.

2. SEM examination methods and results missing analyses to be taken of the images, e.g. morphology, pore sizes, etc.

3. 3D printing methods missing identification of characteristic to be evaluated in results. 

4. Proliferation assay failed to indicate statistical comparison method to be performed between the control or different concentrations.  Further, data unreliable without control (i.e. untreated cells) plot for comparison. Finally, please identify the number of independent determinations used in calculation of the standard deviation. 

5. Rheology time sweep cycle results difficult to visualize in Figure 3.  Consider adding an exploded view or second plot with only fewer cycles or table with key values. Please refer to this publication to cross check recommended data reportable for intended use as shear thinning bioink (see Fig 2): https://pubs.acs.org/doi/10.1021/acsbiomaterials.7b00734

6. Results shown in Figure 6 appear to contradict Figure 4.

7. Missing swelling characteristics and LCST temperature

8. Figure 8 missing scale bar.

Author Response

Reviewer #2

The current manuscript describes an investigation of the synthesis of a novel thermogelling hydrogel and characterization of properties relevant for the application as a bioink. The authors indicate 4 variations of the mixture including 1, 5, 10, and 20% weight by volume, and further characterized the 20 wt%.  

Overall, the investigation has indicated multiple appropriate characterizations to analyze the proposed application.  However, the level of development in methodology and presentation of results is lacking.  

Examples for the level of detail missing in methods includes the following:

1. H-NMR and GPC mentioned performed in methods and discussion, results never presented. 

Response:We thank the reviewer for this important comment. As we answered to reviewer 1, we were quite sure that we also submitted the extensive supporting information with our first submission. If that was not the case, we apologize and of course will provide this upon submission of the revised manuscript.

2. SEM examination methods and results missing analyses to be taken of the images, e.g. morphology, pore sizes, etc.

Response:We thank the reviewer for this comment. However, as we try to point out in the manuscript, we cannot rule out that many details in the presented SEM pictures are probably due to artefacts from the preparation. We therefore would refrain from such more detailed analysis, because we would overemphasize apparent facts which we are not overly confident at this point. 

3. 3D printing methods missing identification of characteristic to be evaluated in results.  

Response:We thank the reviewer for this valuable comment. We added some more information in the materials and methods and in the results discussion (page 15) and believe that this should make the experiments reproducible.

4. Proliferation assay failed to indicate statistical comparison method to be performed between the control or different concentrations.  Further, data unreliable without control (i.e. untreated cells) plot for comparison. Finally, please identify the number of independent determinations used in calculation of the standard deviation.  

Response:We thank the reviewer for this valuable comment. Indeed, all experiments were of course relative to control, as we tried to indicate with the label of the y-axis: relative cell viability. We adjusted the figure caption and added a sentence to the materials and methods section. Also, we added that we conducted the experiment in triplicates. With respect to the statistical comparison, we kindly disagree with the reviewer. We see no point to perform a statistical comparison in the present case, that would be a meaningless and probably misleading exercise.

5. Rheology time sweep cycle results difficult to visualize in Figure 3.  Consider adding an exploded view or second plot with only fewer cycles or table with key values. Please refer to this publication to cross check recommended data reportable for intended use as shear thinning bioink (see Fig 2): https://pubs.acs.org/doi/10.1021/acsbiomaterials.7b00734

Response: We thank the reviewer for this comment, which mirrors a comment from reviewer 1. We added Fig. 3c, where just two cycles are presented. In addition, the suggested reference was added. However, we also decided to leave the values of loss moduli in the Fig. 3c, due to the fact that it is crucial parameter characterizing the amount of the energy dissipated to heat during the mechanical stimulation. In addition, we have added Fig. 5c, which was used for estimation of the yield stress from the steady shear stress mode measurement. Please see the revised manuscript.

6. Results shown in Figure 6 appear to contradict Figure 4. 

Response:We thank the reviewer for this comment.  Unfortunately, the authors cannot see any contradiction between the Fig. 6 and Fig. 4. In both cases, the storage modulus dominates over the loss modulus at low strains (0.1%) and opposite situation is at high strains (10%). Such phenomena is consistent in both presented figures.

7. Missing swelling characteristics and LCST temperature

Response:We thank the reviewer for this important comment. Due to the fact that the presented hydrogel consisting of triblock copolymer showed gelation based on the physical cross-linking the further swelling investigations are not possible, since the hydrogel will dissolve without further additional cross-linking. Also, an LCST temperature cannot be evaluated. As can be seen in Figure 1, the solutions at concentrations below 20 wt.% are turbid at 5 °C and remain turbid upt to 80 °C. Therefore, the gelation is based on enhancement of the physical cross-linking between micelles upon elevated temperature and is not due to an classical LCST type behavior. In fact, this is quite similar to thermogelling hydrogels based PEG-PLGA or PEG-PCL based triblock copolymers, where also neither block shows exhibits an LCST type behavior. We added a short paragraph on page 8 to explain this more clearly 

8. Figure 8 missing scale bar.

Response:We thank the reviewer for this important comment.  The scale bar was added to the figure, please see the revised manuscript.

Reviewer 3 Report

Authors in this manuscript synthesized a ABA triblock copolymers and observed thermal gelation at higher temperature and concentration (20%) of material in water. They have performed rheological characterization at this concentration to investigate the printable bioink application. They have also studied the cell toxicity studies.

Overall, the manuscript is well constructed. Authors have observed the gelation and followed up with characterization of gel behavior. They have emphasized on physical cross-linking. However, the material characterization is incomplete. The phase transition of this ABA triblock copolymer is interesting. I recommend authors to add DSC characterization of copolymer. What is the melting temperature of this copolymer components. Studying micelle formation behavior with a dye solubilization method is also interesting in the context of this manuscript. They have created a amphiphilic copolymer and it should cause micelle formation. These additional studies would help readers significantly to understand the material properties, Tg, sol-gel transition, micelle formation, and micelle aggregation as temperature rise. 

Author Response

Reviewer #3

Authors in this manuscript synthesized a ABA triblock copolymers and observed thermal gelation at higher temperature and concentration (20%) of material in water. They have performed rheological characterization at this concentration to investigate the printable bioink application. They have also studied the cell toxicity studies.

Overall, the manuscript is well constructed. Authors have observed the gelation and followed up with characterization of gel behavior. They have emphasized on physical cross-linking. However, the material characterization is incomplete. 

Response: As mentioned earlier for other comments by the reviewers, we apologize for failing to submit the supporting information.

The phase transition of this ABA triblock copolymer is interesting. I recommend authors to add DSC characterization of copolymer. What is the melting temperature of this copolymer components. 

Response: We thank the reviewer for this important comment. We struggle to see the relevance of the melting point, or glass transition for these polymers, which are properties of the solid state, as they are dissolved in water in the present manuscript.

Studying micelle formation behavior with a dye solubilization method is also interesting in the context of this manuscript. They have created a amphiphilic copolymer and it should cause micelle formation. These additional studies would help readers significantly to understand the material properties, Tg, sol-gel transition, micelle formation, and micelle aggregation as temperature rise. 

Response:We thank the reviewer for this important comment. The critical micelle concentration for this polymer was studied separately and previously published in a preprint. Concurrently, the manuscript is currently being revised. We therefore added a brief comment referring to this work and the corresponding cmc (page 7).

Round 2

Reviewer 1 Report

I recommend the acceptance of the article in JFB. the authors foowed all my recommendations.

Author Response

We thank the reviewer again for taking the time to help to improve our manuscript

Reviewer 2 Report

The current manuscript describes an investigation of the synthesis of a novel thermogelling hydrogel and characterization of properties relevant for the application as a bioink. The authors determine a threshold composition to form the gel as 20%wt of the ABA polymer for future characterization.

Upon second review of the submission, several errors involving the study design have been detected bringing concerns to the abstract, results and conclusion. 

1. the characterization of the identified thermogelling threshold 20wt% is appropriate for the study.  However, the abstract falsely represents characterization of multiple compositions that do not aid in product performance interpretation.  I recommend updating the abstract as well as section 3.3 to succinctly identify the findings from Fig. 1 and choice of composition to move forward with.

2. the rheology study is missing basic fundamental descriptions to understand the experiment including the type of plate used, size of plate, and height of the experiment.  The choice for strain rate should be indicated as the value may be relative to the plate diameter.  For example, if the diameter changed, does the experiment provide the same result? A more relevant experiment for this study would elucidate the time for the gel to reach 80% of the storage value (of which only 2 cycles are needed for display in Fig. 6); this experimental result would quantitatively assist to identify if the bioink composition will hold a shape when printed or flow, as seen when performed for the qualitative assessment in Fig. 8.  Authors may refer to the work in Peak et al. Langmuir 2018 (https://doi.org/10.1021/acs.langmuir.7b02540) for additional information on this characterization.

Figure 4b shows a sinusoidal shift in the data indicating higher order harmonics are present and that the instrument may not be calibrated.  LAOS analysis recommended for authors to confirm.

To assist the authors for future experiments, Figure 5 appears to reveal experimental artifact due to the presence and stickiness/hydrodynamic forces of water, which may have provided a psuedoyield.  It would be useful to modify/increase the ramp rate to confirm the values. 

In LINE285, the term toughness is incorrectly used for the presented experiment, perhaps the authors intended ‘flow profile’.

3. The authors did perform a validated technique to elucidate the cytocompatibility properties for a polymeric construct.  However, for the present study, the study design to evaluate non-gelling concentrations below the threshold is not representative of the proposed product.  It is recommended for the authors to perform a cytocompatibility assay that evaluates the gelled product in its final state to ensure product performance.

Last, in the prior review, multiple requests to update focus of characterization technique (i.e. pore size vs surface characteristics from SEM, etc) were not substantiated.

I believe the work has merit if these issues are resolved in a repeat study.  Overall, the polymer has promise for the proposed application.  However, without appropriate study design, the interpretation of the product characterization cannot be properly depicted.

Author Response

The current manuscript describes an investigation of the synthesis of a novel thermogelling hydrogel and characterization of properties relevant for the application as a bioink. The authors determine a threshold composition to form the gel as 20%wt of the ABA polymer for future characterization.

Upon second review of the submission, several errors involving the study design have been detected bringing concerns to the abstract, results and conclusion. 

 Response:We thank the reviewer for the shown scrutiny and thoroughness.

1. the characterization of the identified thermogelling threshold 20wt% is appropriate for the study.  However, the abstract falsely represents characterization of multiple compositions that do not aid in product performance interpretation.  I recommend updating the abstract as well as section 3.3 to succinctly identify the findings from Fig. 1 and choice of composition to move forward with.

Response:We thank reviewer for this comment. The abstract was modified in order to clearly distinguish that visual observation of thermogelling capability was performed at various compositions (1-20 wt. %) and at broad temperature range (5-80°C), and further 20 wt. % hydrogel was investigated properly. Unfortunately, we do not understand what the reviewer is meaning to say regarding section 3.3.? We do not have a section 3.3 in the manuscript.

2. the rheology study is missing basic fundamental descriptions to understand the experiment including the type of plate used, size of plate, and height of the experiment.  The choice for strain rate should be indicated as the value may be relative to the plate diameter.  For example, if the diameter changed, does the experiment provide the same result? A more relevant experiment for this study would elucidate the time for the gel to reach 80% of the storage value (of which only 2 cycles are needed for display in Fig. 6); this experimental result would quantitatively assist to identify if the bioink composition will hold a shape when printed or flow, as seen when performed for the qualitative assessment in Fig. 8.  Authors may refer to the work in Peak et al. Langmuir 2018 (https://doi.org/10.1021/acs.langmuir.7b02540) for additional information on this characterization.

Response:We thank reviewer for this valuable comment. The fundamental description of the measurement was added (type of geometry, size and gap). Also, the additional information to the steady shear experiment were added. It is well-known that diameter of the measuring geometry significantly influence the absolute values obtained from the rheometer and two individual measurements are incomparable if they were measured at various gaps or geometry parameters.  As we point out in the manuscript and as should be clear from the pictures provided, the bioink in this form does not hold shape. The recovery to 80% storage modulus seems an arbitrary value picked by Peak et al., but we fail to see its particular relevance, as it cannot generally be seen as an indicator for shape fidelity. We would argue that the point when the storage modulus exceeds the loss modulus is more relevant, in combination with the yield point.

Figure 4b shows a sinusoidal shift in the data indicating higher order harmonics are present and that the instrument may not be calibrated.  LAOS analysis recommended for authors to confirm.

Response:We thank reviewer for this valuable comment. The rheometer Anton Paar MCR-301 has internal user calibration every 100 days to check if the geometry constants and tourque transducer constants are still valid. Every two years, the technician from Anton-Paar company check whether the device provide sustainable results and provide professional calibration. If there is sinusoidal shift in the data, this is something for deeper discussion, but we are confident that our equipment is in proper condition, also as judged from many other measurements with other materials analyzed in our lab. However, in the present case, we would rather argue that hydrogel presented in this study is not covalently but physically cross-linked and that in fact the gel structure is weak. Therefore, both viscoelastic moduli are not entirely frequency independent as the gel-like systems with strong internal structure usually are and some fluctuation are possible as a response of the weak physical cross-links on the various applied frequency. The data presented in Figure 4b was measured within the linear viscoelastic region, thus the LAOS measurement seems to be out of scope of this study. No changes made to the manuscript.

To assist the authors for future experiments, Figure 5 appears to reveal experimental artifact due to the presence and stickiness/hydrodynamic forces of water, which may have provided a psuedoyield.  It would be useful to modify/increase the ramp rate to confirm the values. 

Response:We thank the reviewer for pointing this out. We believe that the reviewer refers to Figure 5c(?) which was already removed, as we had the same concerns as the reviewers. Therefore, we used a different analysis, preventing such artifact (Figure 4c) and we will generally follow the reviewers recommendation for various ramp rates to verify the presented rheological data in the Figure 5.

In LINE285, the term toughness is incorrectly used for the presented experiment, perhaps the authors intended ‘flow profile’.

Response:We thank reviewer for this comment. The term toughness was used in the manuscript twice, and we can agree that in both cases, this was not proper. Once, the toughness was substitute by integrity, since the yield stress is more about the state, when material exhibit elastic behaviour (reversible deformation). Secondly, the term toughness was substituted by yield stress which seems to be more suitable for the proper meaning of this sentence.

3. The authors did perform a validated technique to elucidate the cytocompatibility properties for a polymeric construct.  However, for the present study, the study design to evaluate non-gelling concentrations below the threshold is not representative of the proposed product.  It is recommended for the authors to perform a cytocompatibility assay that evaluates the gelled product in its final state to ensure product performance.

Response:We thank the reviewer for this comment. The reviewer is correct, if this material would be used as a bioink as such, we should test it at higher concentration, at least the concentration at which we would intend to print. However, as we note in the manuscript, the material is too weak at this point to exhibit shape fidelity and therefore is not suitable for 3D printing, we are reluctant to test higher concentrations. Changes made to manuscript: Important to note, if the shape fidelity can be improved, cytocompatibility must again be assessed and also conditions relevant for printing will have to be tested.

Last, in the prior review, multiple requests to update focus of characterization technique (i.e. pore size vs surface characteristics from SEM, etc) were not substantiated.

Response:We would like to refer the reviewer to our previous answer. At this point, we cannot rule out, in fact, we must assume, that many details in the presented SEM pictures are probably due to artefacts from the preparation (freeze-drying). To avoid such artefacts with confidence, one would have to conduct cryo-SEM experiments, which are very difficult, expensive and time-consuming to master. We therefore would refrain from such more detailed analysis, because we would overemphasize apparent facts which are likely artefacts. In fact, we would rather remove the SEM images altogether than analyzing in detail which we believe is not reasonable. However, since we do clearly state that the observed morphology can be affected by the preparation method, we would keep the manuscript as is.

I believe the work has merit if these issues are resolved in a repeat study.  Overall, the polymer has promise for the proposed application.  However, without appropriate study design, the interpretation of the product characterization cannot be properly depicted.

Response:We would like thank again the reviewer for the critical feedback which improved the manuscript in our opinion. Even though we did not agree with all points the reviewer brought up, we are confident that the manuscript is no suitable for publication.

Reviewer 3 Report

Authors addressed requested revisions. The manuscript is suitable for publication in it's current form.

Author Response

(The authors gave the same response as above.)
